# Clinical Perspectives and Management of Edema in Chronic Venous Disease—What about Ruscus?

**DOI:** 10.3390/medicines9080041

**Published:** 2022-07-25

**Authors:** Imre Bihari, Jean-Jérôme Guex, Arkadiusz Jawien, Gyozo Szolnoky

**Affiliations:** 1Vascular Surgery Department, St Rokus Clinical Block, Semmelweis University, 1085 Budapest, Hungary; 2European Board of Phlebology, 1040 Brussels, Belgium; jean-jerome.guex@orange.fr; 3Department of Vascular Surgery and Angiology, University Hospital No. 1, Ludwik Rydygier Collegium Medicum in Bydgoszcz, Nicolaus Copernicus University in Torun, 85-094 Tourn, Poland; ajawien@cm.umk.pl; 4Department of Dermatology and Allergology, Albert Szent-Györgyi Medical School, University of Szeged, 6720 Szeged, Hungary; szolnokygyozo@gmail.com

**Keywords:** acid, ascorbic, edema, hesperidin, *Ruscus*, therapeutics, vascular diseases

## Abstract

**Background:** Edema is highly prevalent in patients with cardiovascular disease and is associated with various underlying pathologic conditions, making it challenging for physicians to diagnose and manage. **Methods:** We report on presentations from a virtual symposium at the Annual Meeting of the European Venous Forum (25 June 2021), which examined edema classification within clinical practice, provided guidance on making differential diagnoses and reviewed evidence for the use of the treatment combination of *Ruscus* extract, hesperidin methyl chalcone and vitamin C. **Results:** The understanding of the pathophysiologic mechanisms underlying fluid build-up in chronic venous disease (CVD) is limited. Despite amendments to the classic Starling Principle, discrepancies exist between the theories proposed and real-world evidence. Given the varied disease presentations seen in edema patients, thorough clinical examinations are recommended in order to make a differential diagnosis. The recent CEAP classification update states that edema should be considered a sign of CVD. The combination of *Ruscus* extract, hesperidin methyl chalcone and vitamin C improves venous tone and lymph contractility and reduces macromolecule permeability and inflammation. **Conclusions:** Data from randomized controlled trials support guideline recommendations for the use of *Ruscus* extract, hesperidin methyl chalcone and vitamin C to relieve major CVD-related symptoms and edema.

## 1. Introduction

Venous disorders of the lower extremities are associated with a range of pathologic conditions, many of which present with edema, complicating the process of making a differential diagnosis [1]. Further, though treatment guidelines for chronic venous disease (CVD) are regularly updated, a range of signs and symptoms are associated with edema within the literature, such as “heaviness” and a “feeling of swelling”. This lack of consistency can result in the liberal interpretation of widely used terms and hinders the exchange of helpful clinical information [2].

A number of treatments for edema in CVD are available. One such treatment is Cyclo3^®^Forte (Pierre Fabre, Paris, France), which was first developed in 1959 and is a combination of an extract from the plant *Ruscus aculeatus*, a traditional herbal product used to relieve symptoms of discomfort and the heaviness of legs related to minor venous circulatory disturbance, and, for the symptomatic relief of itching and burning associated with hemorrhoids [3], the flavonoid hesperidin methyl chalcone and vitamin C (*Ruscus*/HMC/Vit C) [4]. Cyclo3^®^Forte is currently used across 29 countries and is recommended as a first-line treatment for the relief of the major symptoms related to CVD, as well as edema [4,5].

The current article describes presentations at a virtual symposium at the Annual Meeting of the European Venous Forum (EVF) on 25 June 2021. These presentations examined the classification of edema within clinical practice and provided guidance for making a differential diagnosis. In addition, clinical trial data are presented on the efficacy of *Ruscus*/HMC/Vit C for patients with CVD with edema, along with guideline recommendations on the use of *Ruscus*/HMC/Vit C extract.

## 2. Pathophysiology of Edema

Edema occurs as a result of capillary filtration exceeding lymphatic drainage, which results in an accumulation of fluid in the interstitial spaces. Edema is a condition often associated with venous insufficiency in the lower extremities. Two predominant forms of edema are pitting and non-pitting. In pitting edema, the application of pressure on the skin leaves an indentation, which is caused by a low concentration of protein in the fluid within the interstitial spaces [6].

## 3. The Original Starling Principle

In 1896, the Starling Principle was presented to explain microvascular fluid exchange and has been used to provide a better understanding of the pathophysiologic mechanism of fluid build-up in edema. Fundamentally, the Starling Principle proposed that fluid movement across the capillary membrane is dependent on the balance between the two opposing forces on either side of the membrane, the hydrostatic pressure gradient and the colloid osmotic pressure (COP) gradient (Figure 1) [7]. The theory supports the concept that fluid extravasation occurs on arterial segmental capillaries, and a large amount of interstitial fluid is reabsorbed across the venous segment of capillaries; meanwhile, any excess fluid is reabsorbed by the lymphatic system. As the oncotic gradient for reabsorption is between the plasma and the interstitial compartment, raising the plasma oncotic pressure should in turn increase fluid reabsorption from the extravascular compartment [7]. The Starling equation allowed for the calculation of the different pressures that affect fluid movement:$$Jv=\kappa \left[\left(P_{c}-Pi\right)-\sigma \left(\pi p-\pi i\right)\right]$$
where *Jv* is the fluid filtration rate, κ is the hydraulic conductance of microvascular walls, *P_c_* is the capillary hydrostatic pressure, *Pi* is the interstitial hydrostatic pressure, *σ* is the reflection coefficient, *πc* is the COP and *πi* is the interstitial oncotic pressure [7,8].

### 3.1. The Revised Starling Principle

In 2012, a major modification to the Starling Principle was proposed. Mesenteric capillaries from frogs and rats were used to demonstrate that when hydrostatic pressure in the capillary fell below plasma oncotic pressure, fluid absorption occurred transiently, as demonstrated by the Starling Principle. However, this finding was not maintained in the steady state, and no absorption was seen when hydrostatic pressure in the capillaries was lower than plasma oncotic pressure [7,9]. The revised Starling Principle was thus developed. Where the previous understanding was that the protein-free fluid was located between the plasma and the interstitial compartment, it is now understood that the oncotic gradient for fluid reabsorption is between the glycocalyx (a membrane-bound biologic macromolecule semipermeable layer) space and the endothelial cell membrane, as shown in Figure 1 [7].

Since this discovery, the revised Starling Principle has been applied to clinical fluid management, and the equation has been updated in line with these findings:$$Jv=Lps\left[\left(P_{c}-Pi\right)-\sigma \left(\pi p-\pi i\right)\right]$$
where, in addition to the previously described symbols, the hydraulic permeability of the capillaries is represented by L*p* and the surface area available for filtration by *S* [10].

### 3.2. Further Considerations on the Extended Starling Principle

Despite the revisions to the original Starling Principle, Hahn et al. determined that several factors required further consideration [11]. The glycocalyx layer was understood to degrade quickly due to inflammation, surgery and ischemia, resulting in the rapid leakage of proteins from the capillary, which would reduce the persistence of infusion fluids within the vessels. Findings from patients and volunteers, however, showed no large capillary leakage of colloid fluid volume and no degradation of the glycocalyx layer, despite the patients experiencing hypervolemia [11].

One of the more perplexing aspects of the revised Starling Principle was the explanation that the glycocalyx was responsible for transcapillary fluid reabsorption in muscles under certain circumstances when the capillary filtration pressure is reduced (e.g., hemorrhage). Though this may be applicable to experimental models, capillary refill and the reabsorption of interstitial fluid at clinically significant volumes have been thoroughly investigated within the literature and have been shown to persist for hours [11]. The “non-absorption rule” of the revised Starling Principle suggests that raising the plasma oncotic pressure does not recruit fluid from the interstitial compartment. Trials in healthy volunteers, however, contradict this proposal. Following an infusion with 20% human albumin, the volunteers had an increase in their plasma volume that was twice that of the infusion amount, as well as an increase in urine output [11].

Another aspect of the revised Starling Principle that requires further investigation is the theory that hypertonic infusions recruit large amounts of intravascular fluid from the glycocalyx layer. During cardiopulmonary bypass, the hydrostatic pressure of the circuit is kept constant while the plasma oncotic pressure is reduced through dilution with crystalloid fluid. If the sub-glycocalyx region is protein-free, no distribution of priming solution (Ringer’s plus mannitol) would occur, but this is not the case. In fact, normal distribution and a half-life of 8 min is seen in this situation [12]. Since these efforts to reproduce the revised Starling Principle in humans have been met with disappointing results, it is important to conduct further research to validate the revised Starling Principle in humans, especially in a clinical setting [11].

### 3.3. The Importance of Colloid Osmotic Pressure

Within the original Starling Principle, the difference between the COP and hydrostatic pressures on alternate sides of the membrane was responsible for the net fluid shift across the membrane [13]. Within the revised Starling Principle, the COP has been determined to have a lesser role, but it does still warrant consideration [10]. Experiments in humans have demonstrated that there are age-related differences in microcirculatory function. For example, in healthy children, the transcapillary COP gradient has been found to increase with age [13]. In addition, in healthy adults, the thoracic COP was found to be significantly higher than the COP in the calf [14], a finding that was not seen in pediatric volunteers [13].

The permeability of the capillary membranes is another important factor that determines fluid movement across capillary membranes and contributes to the formation of edema [9]. Osmotic pressure is determined by the membrane’s selective permeability. Whereas sodium and chloride ions are relatively ineffective osmotic agents, able to move rapidly between the plasma and interstitial spaces, proteins such as albumin as well as hemoglobin are comparatively very effective osmotic compounds, a point that is often overlooked. Proteins tend to be restricted to the plasma and are able to attract water to the plasma compartment from the interstitial fluid, which has a low concentration of proteins; the plasma oncotic pressure or COP is represented by this osmotic pressure [9].

In particular conditions associated with edema, a lower COP has been implicated as an important factor for clinical consideration. In states of hypoalbuminemia, a significantly reduced COP can cause water and solutes to move from the capillaries to the interstitial spaces [9]. Other states that may be associated with changes in COP include left ventricular failure. As ventricular systole weakens, left ventricular filling pressure increases, triggering counter-regulatory events to restore fluid balance. To counterbalance the increase in hydrostatic pressure, a low-protein filtrate passes through the lung capillaries, resulting in a higher COP in the plasma, and the pulmonary lymphatic system clears the excess fluid from the air spaces until this ‘safety valve’ becomes overwhelmed [9]. During normal pregnancy, the COP is reduced as the plasma volume increases until approximately 30 to 34 weeks’ gestation when COP increases again until term [15]. The microvasculature damage that occurs during diabetes can increase the protein permeability of capillaries, which in turn results in COP changes [16].

Another variable that warrants consideration and a high degree of accuracy in the Starling equations is the interstitial hydrostatic pressure (*Pi*). In the skin and muscle, during basal conditions, the *Pi* is approximately −1 mmHg. However, during inflammation within the same tissues, the *Pi* is approximately −10 to −15 mmHg. During acute inflammation, this *Pi* decrease may be counterbalanced by an increase in interstitial fluid volume secondary to albumin leaking from the capillary [17].

A theory regarding the pathophysiologic mechanism of edema formation in CVD suggests that mild alterations in COP and interstitial pressures may increase the propensity for swelling [17,18].

## 4. Edema from a Clinical Perspective

Edema may present as either symptomatic or asymptomatic swelling [19]. Lower leg edema is frequently the first clinical indication of chronic venous insufficiency (CVI); however, it can represent various conditions and can vary in its clinical presentation [6,20]. To make a differential diagnosis, a complete patient history is recommended. The clinical interview should aim to determine if the patient has a history of the condition and also look for possible predisposing factors [21].

The clinical examination should identify whether the edema is isolated or diffuse, painful or not, assess the consistency (pitting or non-pitting) and determine whether there is head and neck involvement or urticaria-like lesions (suggesting angioedema) and signs of thromboembolic disease or internal organ disease [21,22]. For limb edema, it is necessary to assess the symmetry (unilateral or bilateral), evaluate whether the edema is acute (<72 h) or chronic (>3 months) and evaluate the severity of the edema. Both chronic and acute edema may present bilaterally or unilaterally. A common cause of acute leg edema—both unilateral or bilateral—is deep vein thrombosis, whereas chronic edema can present as primary venous disease in unilateral edema or post-thrombotic syndrome in bilateral edema [19]. Whether there are changes in the edema based on the time of day should also be determined [19]. Further, lower limb edema may be the result of arthrogenous stasis syndrome. Restricted mobility due to a disability, for example, prevents the emptying of the veins in the calf muscle and can cause edema in the ankle.

For two common forms of edema, venous and lymphatic, a differential diagnosis may be made based on the clinical criteria. Venous edema presents primarily in the ankle or lower part of the calf (though the foot may also be involved). It is relieved after rest or elevation of the feet, and it worsens during the day. In contrast, lymphatic edema frequently forms a dorsal lump on the foot [19] and can occasionally be identified by the presence of a positive Stemmer sign, which has high sensitivity (92%) but moderate specificity (57%) for advanced primary or secondary lymphatic edema [23]. The diagnostic evaluation for venous edema may be performed with the widely used Duplex ultrasound scan or, alternatively, the functional ambulatory venous pressure test or the air plethysmography test, which measures volume [24]. Lymphatic edema may be measured with lymphoscintigraphy or indocyanine green fluorescent lymphography [25].

The cause of diffuse edema is likely heart failure, liver failure, or a kidney disorder. Edema limited to the lower limbs may be caused by infection, particularly erysipelas in the case of isolated edema, or may instead be of venous origin, lymphedema, lipedema or originate from other causes [21]. Venous edema can be the result of a primary defect in the superficial or deep veins, as well as post-thrombotic diseases, obstacles in venous return (e.g., May–Thurner syndrome), or venous malformation (e.g., Klippel–Trenaunay syndrome) [21].

Medication use may also cause or worsen peripheral edema [26]. Some vasodilators, such as calcium channel blockers, may cause an increase in capillary hydrostatic pressure, which results in extravasation of fluid to the interstitium and edema [27,28]. Non-steroidal anti-inflammatory drugs (NSAIDs) have been found to elevate blood pressure and are associated with edema [29]. NSAIDs inhibit certain cyclo-oxygenases, which are required for the production of prostaglandins, and these are in turn responsible for a range of physiological processes, including the maintenance of renal hemodynamics and tubular reabsorption of water and sodium [30]. Opioids, such as morphine, have also been found to cause peripheral edema secondary to a histamine-modulated release of nitric oxide causing vasodilation [31,32]. The use of certain anti-cancer drugs can increase capillary permeability, with protein-rich fluid moving into the interstitial space, causing edema [33].

## 5. Signs and Symptoms of Edema

Edema should be considered as a both an observable sign (swelling) and a symptom described by the patient (e.g., sensations of heaviness or swelling), even when not obvious upon observation.

In the CEAP classification (described in more detail below), edema is the C3 sign of CVD, but edema is often described in the literature as a symptom and has previously been associated with other non-specific terms such as “heaviness” and “feeling of swelling” [2,20,34]. Such mixed terminology within the literature may complicate the interpretation of research into the reported prevalence and treatment of edema based on the choice of diagnostic tests and assessments.

To measure edema as a symptom of venous disease, both visual and numerical scales are available, as well as quality of life (QoL) questionnaires [35,36]. Edema can be assessed by volumetric measurements, such as water displacement (which measures the amount of water the leg displaces when put inside a container), the more modern optoelectronic volumetry (which calculates the leg volume using infra-red rays) or the digital image three-dimensional modeling technique [20,36]. Alternatively, leg circumference measurements using the spring tape or Leg-O-Meter may be used. As the Leg-O-Meter has a fixed base, it may allow for more accurate measurements, as the circumference of the leg may be measured repeatedly across visits at the same point on the leg, whereas the spring tape only measures the leg circumference, and care is required to ensure the same point is measured each time. Alternatively, parallel measuring tapes may be used, which are similar to the Leg-O-Meter with a fixed base, but instead measure the leg at multiple fixed points across 4 cm intervals and also feature a 20-g weight at the end [20,35].

## 6. The Prevalence of Edema

Due to the discrepancies in the use of terminology associated with edema of the lower extremities, determining its precise prevalence in patients with CVD has been challenging, with limited consistency apparent in the literature [1]. In a multicenter, cross-sectional study by Jawien and colleagues [1], the prevalence of CVI was assessed in 40,095 people visiting a doctor (general practitioner, gynecologist or internist) in Poland. The majority of the participants were female (84%), the mean age was 44.8 years and 10% presented with edema. Of those who were assessed according to the CEAP classification system, 4.5% were reported to have C3 (edema). Comparatively, edema was present in 61.1% of people with varicose veins and 20.1% of those who did not have varicose veins [1]. Symptoms such as leg heaviness and leg aches were more prevalent than edema in both groups. In addition, edema was specifically reported in 56.1% of patients with CVI, whereas 13.3% of people without CVI reported edema. Leg heaviness (a symptom of edema) was reported in 73.7% of patients with CVI and in 23.3% of people in the non-CVI group. These findings indicate that symptoms of venous disease presented more commonly than edema as a sign, suggesting that patients experience symptoms before edema is clinically apparent, highlighting the need for standardization in terms relating to edema. They also support the theory that CVI is likely the cause of the high level of edematous symptoms [1].

The Bonn Vein Study [37] was conducted by the German Society of Phlebology to investigate the prevalence and severity of CVDs in the German urban and rural residential populations. Between 2000–2002, 3072 participants (56% women) who were 18–79 years of age were randomly chosen from the population registers. A history of leg swelling was reported in 16.2% of men and 42.1% of women. Uni- or bilateral leg swelling within the previous 4 weeks was reported in 14.8% of participants, and typical symptoms associated with vascular disorders (within the last 4 weeks) were reported in 56.4%. Of the whole population, 14.2% were classified as C3 according to CEAP (11.6% of men and 14.9% of women) [37], a higher proportion than was seen in the study by Jawien and colleagues [1].

In a cross-sectional, post-hoc analysis of the Bonn Vein Study, the relationship between venous disorders and leg symptoms within urban versus rural settings in Germany was further investigated [38]. Leg symptoms (e.g., swelling, heaviness, tightness, skin irritation, pain and muscle cramps) were assessed using a standardized questionnaire. Of the 2624 participants (48.7% male) assessed, heaviness was reported by 2610, and a feeling of swelling was reported by 2614. When comparing the odds ratio (OR) for urban versus rural living, heaviness was reported as 1.1 (95% confidence interval [CI] 0.9, 1.3), and a feeling of swelling was 1.3 (95% CI 0.9, 1.7). The OR for heaviness in those with versus without varicose veins was 1.5 (95% CI 1.2, 2.0), and the OR for a feeling of swelling was 1.5 (95% CI 1.1, 2.0). Finally, comparing those with and without CVI, the OR for heaviness was 1.6 (95% CI 1.2, 2.1), and for a feeling of swelling it was 3.2 (95% CI 2.4, 4.5). Symptoms of heaviness and the feeling of swelling were significantly less prevalent in patients with CEAP class C0 compared with C2 and above [38]. These researchers found that certain symptoms (itching, feeling of heaviness and tightness) were more closely associated with venous diseases than the symptoms of restless leg or muscle cramps. Consequently, it was suggested that restless leg symptoms and muscle cramps should no longer be considered as venous leg symptoms [38].

Symptoms associated with CVD tend to worsen with warmth and prolonged standing and are more prevalent with an increasing body mass index (BMI) [39].

## 7. A Diagnostic Algorithm for Chronic Lower Extremity Swelling

A recent publication by Gasparis and colleagues has suggested a diagnostic algorithm to assist with the differentiation of chronic edema of the lower extremities (Figure 2) [19].

Patients who present with peripheral edema for >3 months should be evaluated for systemic causes. If no systemic cause has been detected, and no or only partial improvement is seen, a thorough history and clinical evaluation should be conducted before the patient undergoes diagnostic venous imaging (either venous duplex imaging for reflux and obstruction of lower extremities or imaging of the inferior vena cava and iliac veins). Screening specific to the stage of CVD, lymphedema or lipedema should be conducted prior to the final diagnosis [19].

## 8. CEAP Classification

The CEAP classification is an internationally accepted, standardized classification system used to describe the stages of CVD (Table 1), and edema is the C3 sign within CEAP. However, the severity of edema and the degree of induration (soft vs. firm) are not considered in the CEAP classification, and phlebolymphedema is not addressed [40].

To further assist with the effective diagnosis of CVD, a recent publication has presented an updated CEAP classification list (Table 1) [40].

The updated list has maintained the original classes, but has added new sub-classes (C_2_r, C_4_a, C_4_b and C_4_c) [40]. Given the complexity associated with making a differential diagnosis for a patient with edema, it may be prudent to consider the expansion of the edema class to include subcategories of C_3_ during a future revision of the classification system, as the current C_3_ criterion does not specify any causes or quantify the extent of the edema [40].

## 9. The Benefits of *Ruscus* Extract on Edema

The mechanism of action of *Ruscus* extract on venules is multifactorial. First, *Ruscus* extract can increase venous tone through a direct and an indirect α–adrenergic effect and has been demonstrated to improve the contractility of peripheral lymphatic vessels [41]. *Ruscus* extract can also ameliorate the increase in macromolecular permeability that is stimulated by histamine, bradykinin or leukotriene B4. Lastly, *Ruscus* extract has been found to have an anti-inflammatory effect via the inhibition of endothelial cell activation during hypoxia, including reducing the adenosine triphosphate (ATP) concentration, which would otherwise lead to the activation of inflammatory mediators [42]. The pharmacologic formulation of *Ruscus* extract also contains HMC, which, in addition to decreasing capillary permeability, has been found to have various anti-inflammatory effects, including reducing nuclear factor-κB (NF-κB) activity, oxidative stress and cytokine production, and so works synergistically with *Ruscus* extract [43]. The other component, Vit C, also known as ascorbic acid, has been reported to improve capillary resistance and prevent capillary rupture [44].

## 10. Clinical Studies with *Ruscus*/HMC/Vit C

Many current studies continue to address the symptom of the sensation of swelling instead of addressing edema exclusively as a sign [45]. Given the strong link between this particular symptom and the C3 sign [46], the following studies examined the role of *Ruscus* extract in treating edema, as both C3 and the sensation of swelling.

Multiple observational prospective studies have evaluated the effect of *Ruscus*/HMC/Vit C on edema and other signs and symptoms of CVD (Table 2) [47,48,49,50,51,52]. In these studies, patients received two or three capsules of *Ruscus*/HMC/Vit C per day for between 1 and 6 months.

All of these studies showed a reduction in edema, in terms of the proportion of patients affected or as the mean ankle circumference [47,48,49,50,51,52]. Patients also reported reductions in the severity of other symptoms, such as pain or cramps [47,49,50,51,52], and in one study that used plethysmography for objective assessment of venous refilling, there was a significant correlation between symptom improvement and plethysmographic parameters [47]. The studies by Guex and colleagues also evaluated patient QoL using the Short Form 12 (SF-12) and the Chronic Venous Insufficiency Questionnaire (CIVIQ) and found significant improvements in QoL during treatment with *Ruscus*/HMC/Vit C [49,50,51].

A randomized, double-blind, placebo-controlled study in 60 patients in Argentina with uncomplicated CVI assessed the effect of two capsules per day of *Ruscus*/HMC/Vit C over the course of 2 months. Functional symptoms and clinical signs of CVI were measured at baseline, and then after 15, 30 and 60 days. The sensation of evening edema reduced in intensity and was significantly improved from day 15 in the *Ruscus*/HMC/Vit C group compared with the placebo group (*p* = 0.03), and mean ankle circumference decreased significantly in the active treatment group compared with placebo by day 60 (*p* = 0.02) [53].

The above clinical trials demonstrate the value of *Ruscus*/HMC/Vit C across a range of clinical assessments, as well as across different populations.

A systematic review and meta-analysis reviewed the ability of *Ruscus*/HMC/Vit C to improve individual venous symptoms and edema across 10 randomized, double-blind placebo-controlled trials [45]. When assessed as a continuous variable, the standardized mean difference (SMD) between *Ruscus* extract and placebo in the feeling of swelling was −2.27 (95% CI −3.83, −0.70) across three studies with 150 patients. The categorical variable risk ratio (RR) was 0.53 (95% CI 0.4, 0.71) across five studies with 217 participants, and the number needed to treat (NNT) for the feeling of swelling was 4 (95% CI 2.6, 8.0) [45]. Additionally, for patients with leg edema, the SMD between *Ruscus* and placebo for ankle circumference was −0.74 (95% CI −1.01, −0.47) across four studies with 228 patients and for leg or foot volume was −0.61 (95% CI −0.9, −0.31) across three studies with 181 patients. Overall, the evidence provided within the review was deemed to be of a high quality, supporting the conclusion that *Ruscus* extracts are highly effective in reducing edema in patients with CVD [45].

## 11. *Ruscus*/HMC/Vit C in the Chronic Venous Disease Guidelines

Guidelines published in 2018 addressing the management of chronic venous disorders of the lower limbs assessed the relative effectiveness of different venoactive therapies [5]. These guidelines rate the evidence as Grade A (high quality) for data supporting the significant reduction in the feeling of swelling, ankle circumference and foot/leg volume for patients taking *Ruscus*/HMC/Vit C compared with placebo [5], citing the meta-analysis described above [45].

## 12. Conclusions

Edema associated with CVD is a significant problem that can be challenging to diagnose and manage. Current CVD guidelines recommend the use of *Ruscus*/HMC/Vit C to reduce the signs and symptoms associated with edema in patients with CVD based on high-quality clinical trial data across a range of populations.

## Figures and Tables

**Figure 1 medicines-09-00041-f001:**
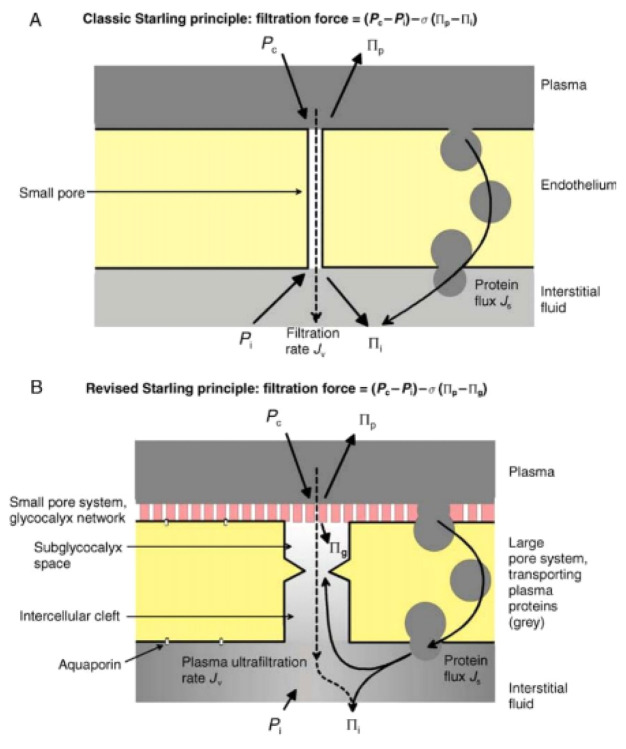
A comparison of the classic and revised views of the Starling Principles showing the forces acting on the endothelial semi-permeable membrane [7]. (**A**) The classic view showing the endothelium as a semipermeable layer. (**B**) The revised view showing the glycocalyx as the semipermeable layer; grey shading represents the protein concentration. *Jv* is the fluid filtration rate, *P_c_* is the capillary hydrostatic pressure, *Pi* is the interstitial hydrostatic pressure, *σ* is the reflection coefficient, *πc* is the capillary oncotic pressure and *πi* is the interstitial oncotic pressure. Republished with permission of Elsevier, from [7] permission conveyed through Copyright Clearance Center, Inc.

**Figure 2 medicines-09-00041-f002:**
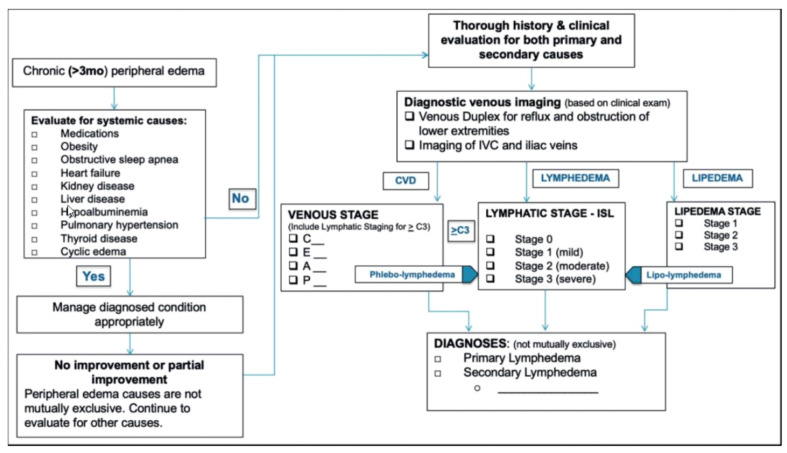
The diagnostic algorithm proposed for the differentiation of chronic lower extremity swelling [19]. IVC, inferior vena cava; C, CEAP classification; CVD, chronic venous disease. From [19]. Reprinted by Permission of SAGE Publications.

**Table 1 medicines-09-00041-t001:** The 2020 revision of CEAP: a summary of clinical classifications for cardiovascular disease [40]. Reprinted from [40], with permission from Elsevier.

	Description
C_0_	No visible or palpable signs of venous disease
C_1_	Telangiectasias or reticular veins
C_2_	Varicose veins
C_2r_	Recurrent varicose veins
C_3_	Edema
C_4_	Changes in skin and subcutaneous tissue secondary to CVD
C_4a_	Pigmentation or eczema
C_4b_	Lipodermatosclerosis or atrophie blanche
C_4c_	Corona phlebectatica
C_5_	Healed venous ulcer
C_6_	Active venous ulcer
C_6r_	Recurrent active venous ulcer

CVD, chronic venous disease.

**Table 2 medicines-09-00041-t002:** Observational, prospective studies examining the effects *Ruscus aculeatus* extract, hesperidin methyl chalcone and vitamin C (*Ruscus*/HMC/Vit C) in patients with chronic venous disease.

Reference	Country	Patients	N	*Ruscus*/HMC/Vit C Dose (Capsules/Day)	Duration	Effect on Edema
de Oca Narváes, et al. 2007 [48]	Mexico	CVI	170	2	6 months	Proportion of patients with edema ↓ from 84% at baseline to 23% at study end
Peralta et al. 2007 [52]	Mexico	CVI	124	2	12 weeks	Proportion of patients with edema ↓ from 82% at baseline to 0% at study end
Guex et al. 2008 [50]	Argentina	CVD (CEAP class C0 to C3)	1036	3	12 weeks	Mean ankle circumference ↓ by 21 mm from baseline (*p* < 0.001)
Guex et al. 2009 [51]	Mexico	CVD (CEAP class C0 to C3)	917	2	12 weeks	Mean ankle circumference ↓ from 247.8 mm at baseline to 234.64 mm at week 12 (*p* < 0.001)
Guex et al. 2010 [49]	Mexico and Argentina	CVD (CEAP class C0s to C3)	1953	2	12 weeks	Sum of left and right mean ankle circumference ↓ from 509.4 mm at baseline to 488.1 at week 12 (*p* < 0.001)
Allaert et al. 2011 [47]	France	CVD (CEAP class C2s to C3s)	65	3	28 days	Overall frequency of edema ↓ from 88% at baseline to 60% on day 28, and evening edema ↓ from 72% at baseline to 52% on day 28

BID, twice daily; CEAP, clinical (C), etiological (E), anatomical (A) and pathophysiological (P); CVD, chronic venous disease; CVI, chronic venous insufficiency; OD, once daily; TID, three times daily.

## Data Availability

No new data were created or analyzed in this study. Data sharing is not applicable to this article.

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
