# Peer review of "Clinical Perspectives and Management of Edema in Chronic Venous Disease—What about Ruscus?"

_medicines, 2022, doi:10.3390/medicines9080041_

Round 1

Reviewer 1 Report

A well written review on venous edema and drugs possibly influencing venous symptoms. The paper is well written and the text is clear and easy to read.

  The main question of the research is the utility of treatment of venous patients with a combination of natural components. The first part of the paper is an interesting review of current theories on venous edema. The combination of components considered seems to be useful for venous patients.  The topic is quite original. Similar studies with other components have been already published. It adds the possible use of the combination they evaluated to the subject area.   The conclusions are consistent with the evidence and arguments presented and the main question is properly addressed.

Author Response

We would like to thank the reviewer for their review of our paper. We are pleased that it meets with their approval. We note that this reviewer also stated that the English language and style of the article were fine with minor spell check required. To address this, a native English speaker has read and edited the manuscript. Please, see enclosed.

Reviewer 2 Report

Rhizoma Rusci is a raditional herbal medicinal product to 1) relieve symptoms of discomfort and heaviness of legs related to minor venous circulatory disturbances. Indication 2)  for symptomatic relief of itching and burning associated with haemorrhoids, after serious conditions have been excluded by a medical doctor.

Please add the relevant data 

https://www.ema.europa.eu/en/documents/herbal-monograph/european-union-herbal-monograph-ruscus-aculeatus-l-rhizoma-revision-1_en.pdf

Author Response

The relevant data from the Ruscus aculeatus herbal monograph has been added to the introduction (paragraph 2) as requested.
